# A New Look at Low-Rank Recurrent Neural Networks

## Abstract

Low-rank recurrent neural networks (RNNs) have recently gained prominence as a framework for understanding how neural systems solve complex cognitive tasks. However, fitting and interpreting these networks remains an important open problem. Here we address this challenge using a perspective from the "neural engineering framework", which shows how to embed an arbitrary ordinary differential equation (ODE) into a low-rank RNN using least-squares regression. Under this perspective, individual neurons in a low-rank RNN provide nonlinear basis functions for representing an ODE of interest. This clarifies limits on the expressivity of low-rank RNNs, such as the fact that with a $\tanh$ non-linearity they can only capture odd-symmetric functions in the absence of per neuron inputs or biases. Building on this framework, we propose a method for finding the smallest low-rank RNN to implement a given dynamical system using a variant of orthogonal matching pursuit. We also show how to use regression-based fitting to obtain low-rank RNNs with time-varying dynamics. This allows for the rapid training of vastly different dynamical systems that nevertheless produce a given time-varying trajectory. Finally, we highlight the usefulness of our framework by comparing to RNNs trained using backprop-through-time on neuroscience-inspired tasks, showing that our method achieves faster and more accurate learning with smaller networks than gradient-based training.

## 1 Introduction

Recurrent neural networks (RNNs) provide a popular tool for analyzing the computational capabilities of neural populations and the mechanisms that enable them to carry out complex cognitive tasks (Miller et al., 2003; Barak, 2017; Mastrogiuseppe & Ostojic, 2018; Schaeffer et al., 2020; Duncker & Sahani, 2021; Dubreuil et al., 2022). A substantial literature focuses on "task-driven" approaches in which an RNN is trained to perform a particular cognitive task of interest, and then analyzed to determine what dynamics it uses to solve the task Mante et al. (2013); Sussillo (2014); Kanitscheider & Fiete (2017); Pollock & Jazayeri (2020); Turner et al. (2021). Specifically, this involves training an RNN to reproduce a target trajectory (or teacher signal) representing the task dynamic and interpreting it to infer the overall underlying flow-field. However, interpretation of such networks is still a noteworthy problem of interest.

A wide variety of methods have been proposed for training such RNNs including: (1) reservoir computing methods, in which a fixed set of random recurrent weights generate a high-dimensional nonlinear dynamic, and training is applied only to output weights (Jaeger, 2001; Maass et al., 2002); (2) FORCE training, in which a set of random recurrent weights are adjusted using iterative low-rank updates via recursive least-squares Sussillo & Abbott (2009); DePasquale et al. (2018); and (3) methods that adjust all recurrent weights using deep-learning inspired approaches such as back-propagation-through-time (BPTT) Pearlmutter (1990); Sussillo & Barak (2013); Lillicrap & Santoro (2019). Although recent literature has focused primarily on this latter class of gradient-based training methods, they face a variety of challenges, including high computational cost, sensitivity to initialization and hyper-parameters (e.g., learning rate, network size), and susceptibility to vanishing and exploding gradients (Lukoševičius & Jaeger, 2009; Schuessler et al., 2020; Langdon & Engel, 2022; Liu et al., 2023).

Even once they are trained (via any of the above methods), interpreting RNNs to gain insight into task performance remains challenging. Common approaches tend to rely on finding fixed or "slow" points and then using dimensionality reduction methods to visualize projected flow fields Sussillo & Barak (2013); Mante et al. (2013). However, fixed-point finding algorithms are difficult to apply to high-dimensional systems, and it often unclear how accurately low-D projections reflect a network's true dynamics. Previous work has shown that task-trained RNNs with similar performance can nevertheless exhibit different dynamics, raising the question of whether their solutions reveal universal features of task computations (Barak, 2017; Williams et al., 2021).

One recent advance that overcomes many of these difficulties is a theory of low-rank RNNs (Mastrogiuseppe & Ostojic, 2018; Beiran et al., 2021; Dubreuil et al., 2022; Valente et al., 2022). Rather than training a high-dimensional network and then attempting to visualize its behavior using low-D projections, this literature has shown that a wide variety of tasks can be implemented directly in RNNs with intrinsically low dimension, where the dimensionality is set by the rank of the recurrent weight matrix plus input dimensions.

A parallel arm of research has focused on developing methods to embed low-dimensional quantities into high-dimensional network activity (Eliasmith & Anderson, 2003; Stewart, 2012; Boerlin et al., 2013; Abbott et al., 2016; Alemi et al., 2018). Of particular relevance to our work is the Neural Engineering Framework (NEF) which determines connection weights between neurons to implement desired dynamics through a structured encoding and decoding process. Specifically, each neuron encodes a stimulus into a specific pattern of action potentials, using what are called basis functions. These functions collectively represent the neural activity observed in response to the stimulus. Next, decoding is achieved through a process of learning a linear weight, optimized via regression, to map the activity back to the original stimulus value. By linking the stimulus to neural activity (encoding) and back to its representation (decoding), NEF effectively derives the weights necessary for neurons to perform complex operations such as integrating inputs, applying nonlinear functions, or generating desired dynamical behaviors. Thus, this approach when applied to target trajectories, like FORCE, teacher-forcing and neural ODE regression frameworks Sussillo & Abbott (2009); Heinonen et al. (2018); Bhat et al. (2020); Hess et al. (2023), also alleviates issues associated with gradient based training (in cases with known dynamics).

In this work, we propose an NEF approach to provide an alternate view to low-rank RNNs which addresses interpretabiltiy issues of such models. Specifically, we develop flexible representations through the construction of a randomized basis spanned by the low-dimensional dynamical system. These basis functions are used as regressors to embed a target non-linear ODE in the low-rank RNN using least-squares regression. Second, using this framework we provide empirical and theoretical evidence regarding the representational limits of low-rank RNNs. In particular, through a geometric interpretation of basis functions, we emphasize the necessity of neuron-specific inputs for embedding odd-symmetric dynamics (a constraint also observed in general RNNs). Third, using a variant of orthogonal matching pursuit we derive the smallest RNN that can implement any target ODE. Fourth, we offer novel insights on the influence of *time-varying* inputs in embedding both autonomous and non-autonomous ODEs. Using this, we provide empirical evidence on validating theories such as, how similar trajectories can arise from significantly different dynamical systems, and underscoring the necessity of perturbations for differentiating between these systems. Lastly, we also apply our method to learn simulated ODEs which arise from a neuroscience binary decision making task, thereby proving the effectiveness of our method.

## 2 BACKGROUND: RECURRENT NEURAL NETWORKS (RNNs)

Consider a population of $d$ rate-based neurons, with membrane potentials $\mathbf{x} = [x_1, \cdots, x_d]^\top$ and firing rates denoted by $\phi(\mathbf{x}) = [\phi(x_1), \ldots, \phi(x_d)]^\top$, where $\phi(\cdot)$ is a scalar function mapping the membrane potential to firing rate (e.g., sigmoid, hyperbolic tangent, or ReLU). The dynamics of a generic RNN are given by the vector ordinary differential equation and a linear output:

$$\dot{\mathbf{x}} = -\mathbf{x} + J\phi(\mathbf{x}) + I\mathbf{u}(t) \tag{1}$$

$$\mathbf{z} = W\phi(\mathbf{x}), \tag{2}$$

where $J$ is a $d \times d$ recurrent weight matrix, $I$ is a $d \times d_{in}$ matrix of input weights, $\mathbf{u}(t)$ is a $d_{in}$-dimensional input signal, and $\dot{\mathbf{x}} = [\frac{dx_1}{dt}, \ldots, \frac{dx_d}{dt}]^\top$ denotes the vector of time derivatives of $\mathbf{x}$. To describe behavioral outputs, the model contains a mapping from network activity to an output variable $\mathbf{z}$: where $W$ denotes a $d_{out} \times d$ matrix of readout weights.

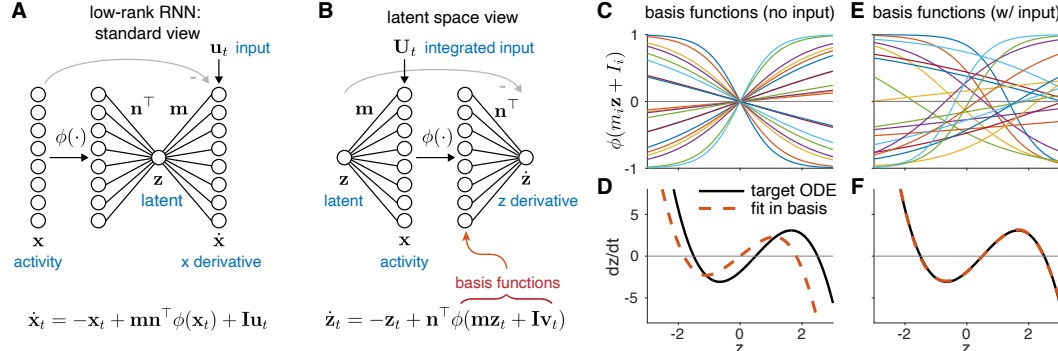

Figure 1: Two equivalent views of low-rank RNNs. **(A)** Standard view of rank-1 RNN with 8 neurons $\mathbf{x}$ and 1 latent dimension $\mathbf{z}$. **(B)** Alternate view of the same network, now framed in terms of the dynamics of latent $\mathbf{z}$. This shows that a low-rank RNN is equivalent to a neural ODE with a single hidden layer (Chen et al., 2018). **(C)** Basis functions obtained by sampling slope parameters $m_i \sim \mathcal{N}(0, 1)$, but without input ($\mathbf{v}_t = 0$). **(D)** Attempting to fit an example ODE using this basis recovers only the odd-symmetric component, since all basis functions are odd symmetric. **(E)** Adding inputs allows basis functions have random horizontal offsets. Here we sampled the input weights $I_i \sim \mathcal{N}(0, 1)$ and set input $\mathbf{v}_t = 1$. (Note that this could also be obtained by using per-neuron "biases"). **(F)** Least squares fitting of $\mathbf{n}$ using the basis from (E) provides good fit to the target ODE.

## 2.1 LOW-RANK RNNs

This network model described above becomes a *low-rank RNN* if the recurrent weight matrix $J$ has reduced rank $r < d$, which implies it can be factorized as:

$$J = MN^\top = \sum_{i=1}^{r} \mathbf{m}_i \mathbf{n}_i^\top. \tag{3}$$

Here $\mathbf{m}_i$ and $\mathbf{n}_i$ represent the columns of the $d \times r$ matrices $M$ and $N$, respectively. In this case, the state vector $\mathbf{x}(t)$ will evolve in a subspace of at most $r + d_{in}$ dimensions (Mastrogiuseppe & Ostojic, 2018; Dubreuil et al., 2022). Activity in the remaining dimensions will decay to zero due to the decay term $(-\mathbf{x})$ in (eq. 1).

In this setting, the network state $\mathbf{x}$ can be re-written as

$$\mathbf{x}(t) = M\boldsymbol{\kappa}(t) + I\mathbf{v}(t), \tag{4}$$

where $\boldsymbol{\kappa}$ is a so-called "latent" vector representing activity in the $r$-dimensional recurrent subspace, and $\mathbf{v}(t)$ represents the low-pass filtered input signals $\mathbf{u}(t)$ Dubreuil et al. (2022); Valente et al. (2022). Finally, the low-dimensional recurrent dynamics can be represented in terms of a differential equation: $\dot{\boldsymbol{\kappa}} = F(\boldsymbol{\kappa}, \mathbf{u})$, where $F$ is a nonlinear function of the latent state $\boldsymbol{\kappa}$ and input $\mathbf{u}$.

## 3 AN ALTERNATE VIEW OF LOW-RANK RNNs

In the framework described above, training a low-rank RNN to produce a desired output $\mathbf{z}(t)$ from an input $\mathbf{u}(t)$ requires learning the model parameters $\{N, M, I, W\}$, which is typically carried out using back-propagation through time Valente et al. (2022). Here we present a different approach to low-rank RNNs, which provides a more intuitive portrait of the network's dynamical capabilities.

We begin by considering the problem of embedding an arbitrary low-dimensional dynamical system into a low-rank RNN. Specifically, suppose we wish to set the model parameters so that $\mathbf{z}$ obeys the dynamics of an particular nonlinear ODE:

$$\dot{\mathbf{z}} = g(\mathbf{z}), \tag{5}$$

for some function $g$. We will then identify this output with the latent vector defining the network's activity in the recurrent subspace: $\mathbf{z}(t) \triangleq \boldsymbol{\kappa}(t)$. This implies that the dimensionality of the output is

equal to the rank of the network, $r = d_{out}$, and constrains the output weights to be the projection operator onto the column space of $M$, that is, $W = M(M^\top M)^{-1}$. We are then left with the problem of setting the weights $M$, $N$, and $I$ so that the latent vector, which we now refer to as $\mathbf{z}(t)$, evolves according to (eq. 5). (Note however the model output need match the dimensionality of the latent variable; in cases where desired output is lower-dimensional than $\mathbf{z}$, the output can be expressed as $\mathbf{z}' = A\mathbf{z}$, where $A$ is some fixed matrix of output weights that, for example, selects only one component of $\mathbf{z}$.)

For simplicity, we begin with the case of a scalar $\mathbf{z}$. This corresponds to an RNN with rank-1 recurrent weight matrix $J = \mathbf{m}\mathbf{n}^\top$, which is simply an outer product of weight vectors $\mathbf{m}$ and $\mathbf{n}$. Assume that the input is also scalar, and that the input vector $I$ is orthogonal to $\mathbf{m}$ (although we can relax this constraint later). Following previous work (Beiran et al., 2021; Dubreuil et al., 2022; Valente et al., 2022) (eq. 4), the network state can be decomposed as a time-varying linear combination of $\mathbf{m}$ and $I$:

$$\mathbf{x}(t) = \mathbf{m}\mathbf{z}(t) + I\mathbf{v}(t), \tag{6}$$

where $\mathbf{v}(t)$ represents the low-pass filtered input, resulting from the linear dynamical system $\dot{\mathbf{v}} = -\mathbf{v} + \mathbf{u}(t)$. The fact that $\mathbf{m}$ and $I$ are orthogonal means that we can write the dynamics governing the latent variable explicitly as:

$$\dot{\mathbf{z}} = -\mathbf{z} + \mathbf{n}^\top \phi(\mathbf{m}\mathbf{z} + I\mathbf{v}(t)) \tag{7}$$

a result shown previously in Valente et al. (2022), and which is schematized in Fig. 1. Given this expression, our goal of embedding an arbitrary ODE $\dot{\mathbf{z}} = g(\mathbf{z})$ into the network can be viewed as setting the model parameters so that

$$g(\mathbf{z}) + \mathbf{z} \approx \mathbf{n}^\top \phi(\mathbf{m}\mathbf{z} + I\mathbf{v}(t)) \tag{8}$$

To achieve this, note that the right-hand-side can be viewed as a linear combination of terms $\phi(m_i\mathbf{z} + I_i\mathbf{v}(t))$ with weights $n_i$, for $i \in \{1, \ldots, d\}$. Each of these terms can be viewed as a basis function in $\mathbf{z}$ for representing the target $g(\mathbf{z}) + \mathbf{z}$. More specifically, if $\phi$ is the hyperbolic tangent function, each such term is a shifted, scaled $\tanh$ function in $\mathbf{z}$, where $m_i$ is the slope and $I_i v(t)$ is the offset. This means that we can view the problem of embedding $g(\mathbf{z})$ into a low-rank RNN as the problem of setting $\mathbf{m}$ and $I$ to build an appropriate set of basis functions, and setting $\mathbf{n}$ so that the linear combination of basis functions approximates $g(\mathbf{z}) + \mathbf{z}$. This approach formalizes the connection between low-rank RNNs and the NEF (Eliasmith & Anderson, 2003; Barak & Romani, 2021), and shows that a low-rank RNN corresponds to a neural ODE with a single hidden layer (Chen et al., 2018; Pellegrino et al., 2023; Pals et al., 2024).

Already, this perspective makes an important limitation clear: if the inputs $\mathbf{v}(t)$ are zero, the basis functions are all odd-symmetric (that is, $g(m_i\mathbf{z}) = -g(-m_i\mathbf{z})$ for all $\mathbf{z}$), crossing the origin only at zero. (see Fig. 1C). Because $-\mathbf{z}$ is also odd-symmetric, and the linear combination of odd-symmetric functions is odd-symmetric, this means that in the absence of inputs, the network can only capture the odd-symmetric component of $g(\mathbf{z})$. A low-rank RNN is therefore not a universal approximator unless it has inputs, or equivalently, different biases or offsets to each neuron (similar to general RNNs). If the $\phi$ is instead taken to be ReLU, the problem is even more severe: each basis function is a linear function with non-zero slope on either $\mathbf{z} > 0$ or $\mathbf{z} < 0$. Thus the network can only approximate $g(\mathbf{z})$ that are piecewise linear functions broken at the origin. (See SI Fig. SI-1).

If we set the filtered input to be the constant $\mathbf{v} = 1$, we see that the problem of embedding an arbitrary ODE in a low-rank RNN amounts to fitting the ODE in a basis of shifted and scaled basis functions in $\mathbf{z}$. To achieve this, we propose to sample the scales (elements of $\mathbf{m}$) and offsets (elements of $I$) to obtain a random basis, and then fit $\mathbf{n}$ by least-squares regression, namely:

$$\hat{\mathbf{n}} = (\phi(\mathbf{z}_{grid}\mathbf{m}^\top + I^\top)^\top \phi(\mathbf{z}_{grid}\mathbf{m}^\top + I^\top))^{-1} \phi(\mathbf{z}_{grid}\mathbf{m}^\top + I^\top)^\top \left(g(\mathbf{z}_{grid}) + \mathbf{z}_{grid}\right), \tag{9}$$

where $\mathbf{z}_{grid}$ denotes a grid of points at which we wish to fit $g(\mathbf{z})$. Note that we could use weighted least squares if we care more about accurately approximating certain regions of $g(\mathbf{z})$, or add a small ridge penalty if the design matrix (whose columns are given by the basis functions evaluated at $\mathbf{z}_{grid}$) is ill-conditioned.

Fig. 1 shows an illustration of this approach for an example ODE, here chosen to be a cubic polynomial with two stable fixed points and one unstable fixed point. Note that the network cannot approximate $g(\mathbf{z})$ when the inputs are set to zero (Fig. 1C-D), but can do so with near-perfect accuracy when both the $\mathbf{m}$ vector and the (constant) inputs $I\mathbf{v}$ are drawn from a Gaussian distribution (Fig. 1E-F)).

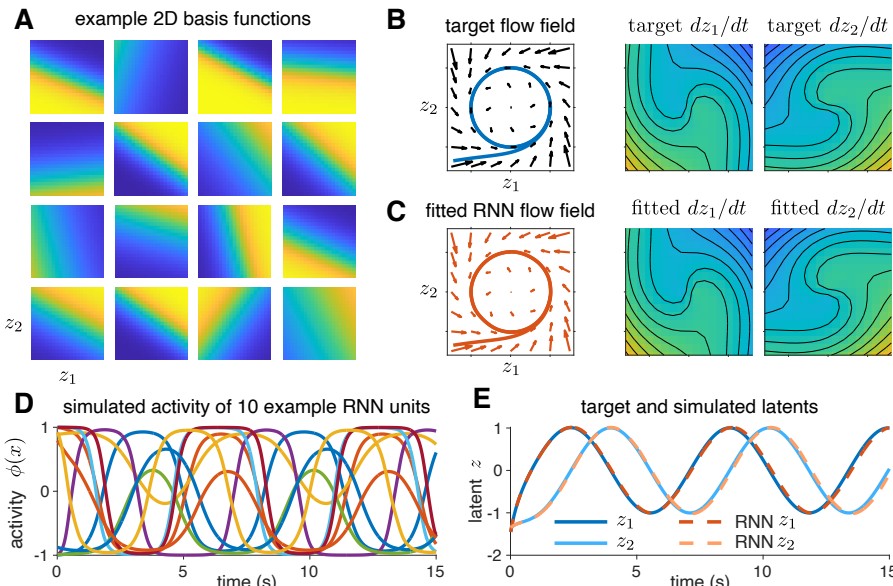

Figure 2: Embedding a 2-dimensional nonlinear ODE into a rank-2 RNN. **(A)** Example basis functions obtained by sampling $M$ and $I$ coefficients from a zero-mean Gaussian, producing randomly oriented, scaled, and shifted hyperbolic tangent functions. **(B)** A target two-dimensional nonlinear dynamical system, containing a stable limit cycle on a circle of radius one, represented as a flow field (left), or by its component functions $g_1(\mathbf{z}) = \frac{dz_1}{dt}$ and $g_2(\mathbf{z}) = \frac{dz_2}{dt}$ (right). **(C)** Least squares fitting of weight vectors $\mathbf{n}_1$ and $\mathbf{n}_2$ produces a near perfect match to the target flow field, and functions $g_1$ and $g_2$. **(D)** Output firing rates $\phi(x_i)$ for 10 example units (i.e $i \in \{1, \dots, 10\}$) during the red example trajectory shown in panel C. **(E)** Simulated trajectories from the true ODE (blue trace in panel B) and latent variable of the fitted RNN (red trace from panel C), plotted as a function of time, showing good agreement between the target ODE and the RNN output. Note that fitting was closed-form, and did not require backprop-through-time.

### 3.1 MULTI-DIMENSIONAL DYNAMICAL SYSTEMS

We can apply this same regression-based approach to higher-dimensional nonlinear dynamical systems, where rank $r = \dim(\mathbf{z}) > 1$. In two dimensions, the basis functions are given by $\phi(m_{1i}z_1 + m_{2i}z_2 + I_i)$, which are scaled, shifted $\tanh$ functions with a random orientation (see Fig. 2A). Approximating a 2D dynamical system with a rank-2 RNN can then be written as the problem of fitting two different nonlinear functions $g_1(\mathbf{z})$ and $g_2(\mathbf{z})$ using two different linear combinations of the same 2D basis functions:

$$g(\mathbf{z}) = \begin{bmatrix} g_1(\mathbf{z}) \\ g_2(\mathbf{z}) \end{bmatrix} \approx - \begin{bmatrix} z_1 \\ z_2 \end{bmatrix} + \begin{bmatrix} \mathbf{n}_1^\top \phi(M\mathbf{z} + I) \\ \mathbf{n}_2^\top \phi(M\mathbf{z} + I) \end{bmatrix}, \tag{10}$$

where $M = [\mathbf{m}_1 \mathbf{m}_2]$ is a $d \times 2$ matrix whose columns define the slope and orientation of each basis function, $I$ is once again a column vector of offsets, and we have assumed constant input ($\mathbf{v} = 1$). Note once again that if we do not include inputs, the basis functions are all radially odd-symmetric around the origin. Thus, once again, the RNN will only be able to capture radially odd-symmetric $g(\mathbf{z})$, and is not a universal approximator unless we include nonzero offsets $I\mathbf{v} \neq 0$.

To embed a given multi-dimensional ODE $g(\mathbf{z})$ into a low-rank RNN, we once again generate a random basis by sampling the elements of $M \in \mathbb{R}^{d \times 2}$ and $I \in \mathbb{R}^d$ from a Gaussian distribution. The problem factorizes into learning each column vector $\mathbf{n}_i$ for each dimension of the $g$, we have:

$$\hat{\mathbf{n}}_i = (\phi(Z_{grid}M^\top + I^\top)^\top \phi(Z_{grid}M^\top + I^\top))^{-1} \phi(Z_{grid}M^\top + I^\top)^\top \big(g(Z_{grid}) + Z_{grid}\big), \tag{11}$$

for $i = 1, 2$. This differs from the 1D case above only in that $Z_{grid}$ is now a $r$-column matrix of grid points, where each row contains the coordinates of a single point in $\mathbf{z}$. Note that these grid

points need not be uniformly sampled; we could sample them from an arbitrary distribution, or use a collection of points from simulating the ODE from a variety of starting points.

Fig. 2 shows an application to an example 2-dimensional nonlinear ODE, in this case containing a stable limit cycle. Note that this 2D system is highly nonlinear and not radially odd-symmetric, so once again (Section A.2), embedding the system in a low-rank RNN fails if we do not include inputs (or per-neuron biases).

## 4 FINDING THE SMALLEST RNN FOR A GIVEN DYNAMICAL SYSTEM

We now turn to the question of finding the smallest RNN that can accurately implement a known nonlinear dynamical system, $g(\mathbf{z})$, focusing on a scalar $z$. In other words we aim to determine the minimum number of neurons ($d' << d$) needed by the network to approximate the function $g(\mathbf{z})$. Our goal is thus to solve an optimization problem that imposes a sparsity constraint on the dimensionality of our basis set $B(\mathbf{m}, I)$, while still solving for an appropriate linear weighting $\mathbf{n}$.

More formally, this implies selecting the best $d'$ entries from $B(\mathbf{m}, I)$, to create a basis:

$$B_{d'}(\mathbf{m}, I) = \begin{bmatrix} \phi(m_{i_1}\mathbf{z} + I_{i_1}) \\ \phi(m_{i_2}\mathbf{z} + I_{i_2}) \\ \vdots \\ \phi(m_{i_{d'}}\mathbf{z} + I_{i_{d'}}) \end{bmatrix}, \quad \text{where } \{i_1, i_2, \ldots, i_{d'}\} \subseteq \{1, 2, \ldots, d\}.$$

Then, a linear weighting $[n_0\ \mathbf{n}']$ is learned using least squares regression, where:

$$g(\mathbf{z}) \approx -n_0 * z + {\mathbf{n}'}^{\top} B_{d'}(\mathbf{m}, I) \quad \text{where } \{\mathbf{n}' = 1 \times d' \text{ vector}\} \tag{12}$$

To achieve the desired optimization of approximating $g(z)$, we begin with a large enough $B(\mathbf{m}, I)$ obtained by sampling from a uniform grid of values for $\mathbf{m}, I$. We then follow an iterative approach, wherein at each iteration $t$, we greedily pick a basis function $i_j$ that has the highest alignment to the current residual estimate of $g(\mathbf{z})$. This can be observed as an adaptation of the well-established orthogonal matching pursuit (OMP) framework. A more detailed description of this process is provided below in Algorithm 1. It is worth noting, changing the original basis set $B(\mathbf{m}, I)$ to $B'(\mathbf{m}, I)$, could result in the algorithm converging to a different minima (global minima in $B'(\mathbf{m}, I)$ could be different from global minima in $B(\mathbf{m}, I)$). However, if the basis sets are equivalent we observe similar performance across simulations (Fig SI-9).

---
**Algorithm 1** OMP for finding smallest RNN
---
1: Select a grid of values $\mathbf{z}$.
2: Create global basis set $\boldsymbol{B}$ using uniformly sampled $\mathbf{m}, I$ values.
3: Initialize $n$ weights using linear decay term only: $n_0 = -(\mathbf{z}^{\top}\mathbf{z})^{-1}\mathbf{z}^{\top}g(\mathbf{z})$.
4: Initialize residual: $r_0 = g(\mathbf{z}) - n_0 * (-\mathbf{z})$
5: At each iteration $t$:

    1. Find basis vector with highest correlation with residual: $i_t = \arg\max_i ||\mathbf{B}_i^T \mathbf{r}_{t-1}|||$
    2. Add new entry to the solution basis, $B_t \leftarrow \boldsymbol{B}_i$
    3. Solve to find new linear weights $\mathbf{n}'_t$ using Eqn 9
    4. Compute the updated residual: $r_t = g(\mathbf{z}) - {\mathbf{n}'_t}^{\top} B_t$
    5. Check for termination based on a predefined sparsity threshold $d' = \text{len}(B_t)$
---

In Fig 3, we apply this method to a simulated 1D ODE, with two stable fixed points and one unstable fixed point. The first row shows the greedily-added basis functions, multiplied by their corresponding learned linear weightings. The bottom row shows their linear combination against the true underlying ODE. Through this iterative process, we observe with just 5 neurons our network almost perfectly reconstructs the target ODE.

It is worth noting, previous work (e.g., Luo et al. (2023); Valente et al. (2022)) have similar dynamics which are learned using BPTT with much larger networks (typically 512 neurons). Our method instead provides an empirical framework to find the minimum number of neurons needed to fit dynamics within estimated margins of error. Furthermore, we believe this could be used to provide insights on hyper-parameter values (such as number of neurons) for neural ODE models or other artificial networks.

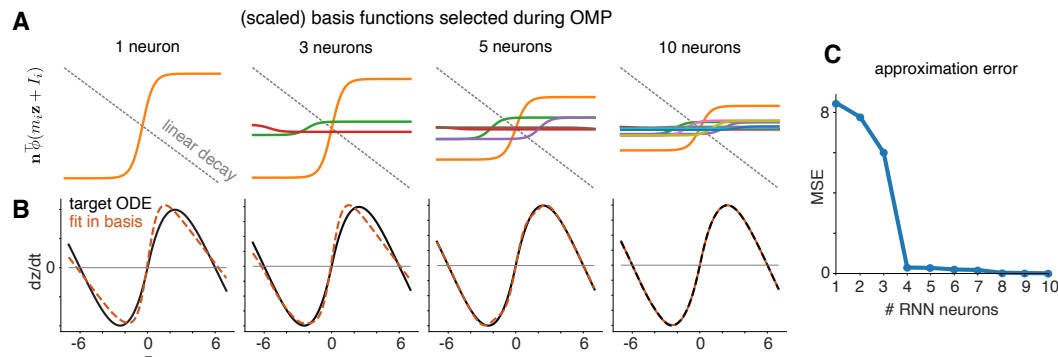

Figure 3: Finding the smallest RNN for a particular nonlinear dynamical system using orthogonal matching pursuit (OMP). **(A)** Scaled basis functions selected after 1, 3, 5, and 10 iterations of OMP, along with the linear decay term $-x$ for an example ODE (shown below). **(B)** Target ODE (black) and RNN fit after each step of OMP. **(C)** Mean squared error (MSE) between target ODE and RNN approximation as a function of the number of RNN neurons added by OMP.

## 5 NEW INSIGHTS INTO THE ROLE OF INPUT DYNAMICS

In Sec. 3, we demonstrated the importance of the presence of inputs in basis functions, $B(\mathbf{m}, I) = \phi(\mathbf{mz} + I\mathbf{v}(t))$. Now, we highlight the influence of the type of inputs (constant or time-varying), in representing arbitrary ODEs. Specifically, our framework so far shows how autonomous ODEs ($g(\mathbf{z})$) are embedded via constant filtered inputs represented as $v = 1$, that do not evolve over time. Here we extend our framework to embed non-autonomous ODEs ($g(\mathbf{z}, t)$) by introducing a dynamical system that governs the evolution of $\mathbf{v}(t)$.

Recent work has suggested an identifiability issue in uncovering underlying low-dimensional dynamics from high-dimensional neural activity, however the role inputs play in this hasn't been explored (Turner et al., 2021; Langdon & Engel, 2022; Liu et al., 2023; Qian et al., 2024). Furthermore, previous work on time varying inputs (Rajan et al., 2016; Remington et al., 2018; Galgali et al., 2023a) has focused primarily on whether observed trajectories were *input driven* or *dynamics driven* . We note here that this latter category can be subdivided into cases where the dynamics themselves are fixed in time or fluctuating due to inputs.

In Fig. 4, we illustrate using our framework to train low-rank RNNs that produce an identical trajectories (blue, red lines in the bottom row) through vastly different dynamical systems (quiver plots in bottom row). In all cases shown, the input dimension $I$ is orthogonal to the dynamics subspace $\mathbf{m}$. Specifically, we demonstrate the following cases:

**1. Autonomous ODE**: Following the general treatment in Sec. 3 above, we embed an ODE of the form $\dot{\mathbf{z}} = g(\mathbf{z})$. The quiver plot (bottom row of panel A) represents the autonomous ODE flow-field illustrated through arrows of the same length across time, for each value of $\mathbf{z}$. Note, this embedding is achieved through inputs that are fixed in time ($\mathbf{v} = 1$). Additionally in the bottom row of panel A, the true trajectory (blue - computed using Euler integration), and the low-rank RNN (red - trained using Eqn 9), both start at the unstable fixed point ($\mathbf{z} = -6$), and eventually converge to the stable fixed point ($\mathbf{z} = +6$). The almost perfect overlap indicates the efficacy of our method.

**2. Non-Autonomous ODE**: In panels B,C,D we embed *time-varying* dynamics of various kinds into different low-rank RNNs. Our training objective was to create time-varying flow fields that produce a trajectory identical to the autonomous ODE case. The existence of time-varying dynamics can be observed by noting arrows of different lengths across time bins (i.e each row of quiver plot).

We first discuss how to embed such dynamics into the low-rank RNN. Intuitively, because the dynamic being approximated represents a different function over time, the basis functions also need to evolve in time. To achieve this embedding, we modeled inputs $\mathbf{v}(t)$ with an exponential decay given by $\dot{\mathbf{v}} = -\mathbf{v}$. This decay results in time-varying basis functions, as different $\mathbf{v}(t)$ values are computed at each Euler time-step when defining $\phi(\mathbf{mz} + I\mathbf{v}(t))$.

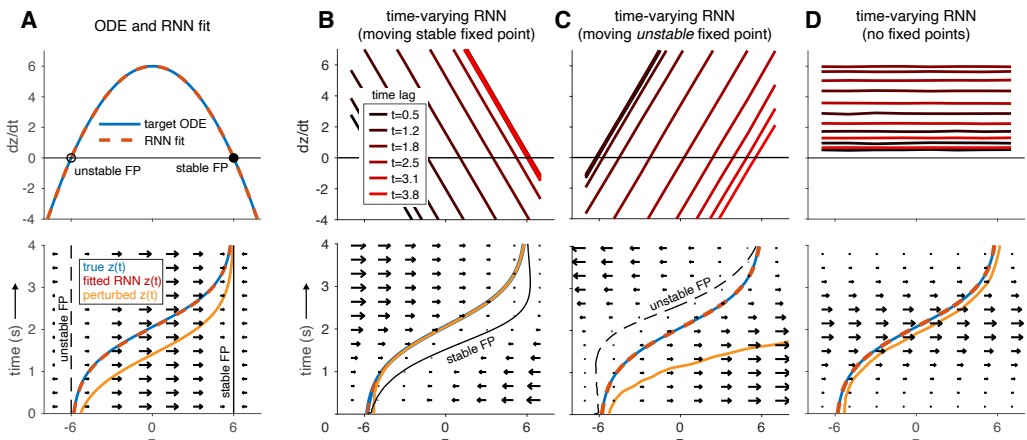

Figure 4: Creating indistinguishable time-varying dynamical systems with time-varying inputs. We show four different time-varying dynamical systems that give rise to the same (blue) target trajectory. **(A)** Top: we defined a target ODE defined by an inverted quadratic function (top), then embedded into an RNN with constant input (following the methods in Sec. 3). Bottom: we then simulated a trajectory that starts just right of the unstable fixed point and converges to the stable fixed point. Here the flow field dynamics do not vary in time. A perturbed input into the RNN (orange) follows the same trajectory, but shifted downward (earlier) in time. **(B)** We trained an RNN to produce a dynamic flow field using time varying basis functions, whose offsets shifted in time after clamping on an input **u** at time zero. Here we set the target to be a time-varying linear dynamical system with a single stable fixed point that moved from left to right. Note that the RNN latent (red) still follows the target trajectory (blue). However, the same perturbation shown in A now converges back to the target trajectory. **(C)** Network trained to produce the same target trajectory using a moving *unstable* fixed point. **(D)** Network trained to produce the same target trajectory using no fixed points. Note however that the trajectory arising from a perturbed initial point (yellow) varies wildly across these models.

Below, we expand on the specific dynamical system governing ODEs that can be approximated using such time-varying basis functions.

1. **Moving fixed points**: As already highlighted, the dynamical system of the target ODE is of the form $\dot{\mathbf{z}} = g(\mathbf{z}, t)$. Panels B, C present specific cases, where this dynamic represents an underlying flow-field which arises due to the diffusion of a single stable or unstable fixed point.
   In other words, our target ODE represents the trajectory of a moving fixed point, modelled through a linear dynamic term in **z** -

$$\dot{\mathbf{z}} = g(\mathbf{z}(t') - \alpha * (\mathbf{z}' - \mathbf{z}(t')) - \mathbf{z}' \qquad (13)$$

   The above equation clearly highlights the systems' state depends on both $\{\mathbf{z}', t'\}$, representing a coarse spacing and a corresponding Euler time bin respectively. First, the term $\alpha * (\mathbf{z}' - \mathbf{z}(t'))$ introduces a linear time-varying correction which moves the fixed point of this system $\mathbf{z}'$ to $\mathbf{z}(t')$ with a rate $\alpha$, as shown in the top rows. Second, $\alpha$ must be positive for panel B (to move a single *stable* fixed point), and negative for panel C (to move a single *unstable* fixed point). Third, while the fixed point dynamic described above is linear, the system still evolves non-linearly due to the function $g$, as shown by the quiver plots. Lastly, in the absence of these dynamics the system decays to the fixed point defined by $\mathbf{z}'$.

2. **No fixed points**: Similar to the moving fixed point case, we can define a system with no fixed points during the movement from -6 to +6. In this case, the target ODE is defined by the non-linear evolution of $\mathbf{z}(t)$ with a time-invariant shift given as

$$\dot{\mathbf{z}} = g(\mathbf{z}(t')) - \mathbf{z}' \qquad (14)$$

In the bottom row of panels B,C,D the true trajectory (blue) and RNN approximated trajectory (red) overlap. Critically, they are all identical, and in fact match the trajectories observed in the bottom row

of panel A. However, as discussed above, they all represent dramatically different target flow-fields, also showcased via their quiver plots.

Our results provide insight into how non-autonomous dynamics can be embedded into low-rank RNNs through time-varying inputs. It is worth noting that most current methods typically interpret low-dimensional dynamics post-training. One specific method that has gained popularity is the zero-finding method, which relies on finding fixed points (Sussillo & Barak, 2013; Smith et al., 2021). Our discussion on non-autonomous dynamics suggests, how similar trajectories can be observed in the presence of dynamical fixed points/ no fixed points at all. This presents a possible failure mode for such techniques. Additionally, we consider the effects of perturbations in the initial point (orange line in bottom row) for these different systems. As observed, while the learned RNN trajectory (black) is identical across all panels, small perturbations in the initial point lead to wildly different trajectories in different models, validating that observations of fixed trajectories are generally not sufficient to uniquely identify the underlying dynamical system Galgali et al. (2023b).

## 6 COMPARISON WITH BPTT FOR NEUROSCIENCE TASKS

Finally, we apply our framework on two neuroscience inspired tasks (Wong & Wang, 2006; Sussillo & Barak, 2013; Luo et al., 2023) and compare our networks against RNN models trained with backprop-through time. Specifically, we implement:

1. **3-bit flip-flop task**: The network receives a series of bits (-1 or +1) in 3 registers, and must store the polarity of the most recent bit in each register (Sussillo & Barak, 2013).

2. **Binary decision-making**: Following previous literature, we model a sensory evidence accumulation task using bi-stable attractors (Wong & Wang, 2006). Sensory inputs drive the system from an unstable fixed point towards one of two stable fixed points, each associated with a different choice (see Appendix B for details).

In Sec. 4, we find the smallest low-rank ($r = 1$) network ($N = 10$ neurons) that almost perfectly fits a bi-stable attractor ODE. Here, we compare our model against networks of the same size (and rank) trained using BPTT (both trained and tested against a set of trajectories simulated from task specific ODEs). As shown, our method provides a closed form solution in one step, as compared to networks trained with BPTT which take an order of one/two more magnitudes to converge. In Fig. 5, panels B,C depict loss curves across a wide range of RNN models trained with BPTT. Consistent with previous findings we observe lower training loss for larger networks, although they take longer to converge. Additionally, low-rank networks ($r = 1$, panel C) have comparable performance for tasks with lower dimensionality. Second, our method qualitatively has better reconstruction of test trajectories (panel A). More importantly, our model significantly outperforms networks (order of magnitude lower test mean-squared error) of the same size (and rank) on a set of test target trajectories (panel D), thereby proving the efficacy of our method.

## 7 DISCUSSION

In this paper, we present an alternative view on low-rank RNNs, that emphasizes interpretability and highlights the representational capacity of such networks. Specifically, we present an NEF inspired framework that directly models the low-dimensional activity and transforms to neural activity space via a fixed linear projection, which is learnt using linear regression (a result also highlighted in Beiran et al. (2021)). Thus, similar to vector-field regression or teacher forcing algorithms Heinonen et al. (2018); Bhat et al. (2020); Hess et al. (2023) our method also overcomes gradient-based training issues (in cases with known/estimated ODEs), with additional advancements described below.

The focus of our work lies in defining a randomized basis which is used to embed an arbitrary non-linear dynamical system. Specifically, through a geometric lens, we show how a low-rank RNN constructs an ODE, as a weighted sum of shifted and scaled basis functions. This makes clear the role that each neuron (single basis function) plays in the low-rank RNN, and also makes evident that inputs are essential for capturing odd-symmetric functions. Thus, with the added advantage of being highly interpretable, our results also clarify when such models (similar to RNNs) behave as universal function approximators (also highlighted in Beiran et al. (2021)).

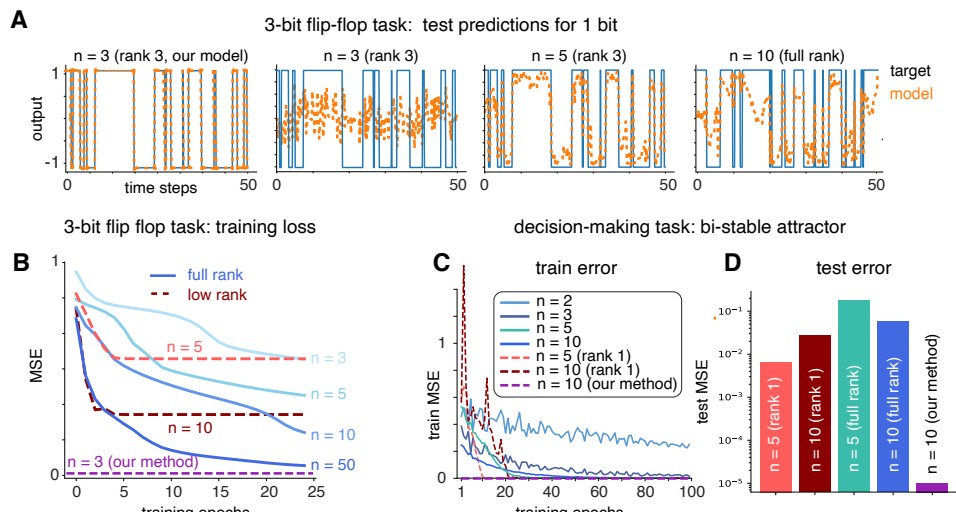

Figure 5: Comparisons to RNNs trained using backprop-through-time (BPTT) for neuroscience inspired tasks. **(A)** Target and trained RNN model outputs for one bit in the "3-bit flip-flop task". Our model (left) achieves near perfect accuracy using only 3 neurons and a rank-3 weight matrix; low rank and full rank networks trained using BPTT do not achieve nearly the same level of accuracy. **(B)** Training loss as a function of number of training epochs for different networks. (Our network, which is trained in one step using least-squares regression is shown in purple for comparison). **(C,D)** Train and test error for RNNs trained to perform a binary decision-making task using bi-stable attractor dynamics (Wong & Wang, 2006; Luo et al., 2023).

Another exciting finding of our work is using OMP to approximate the smallest RNN (from a basis set), which could be used to drive insights on hyper-parameter values for such models. Together, this presents a promising future direction for studying how perturbing connectivity relates to unstudied behavioral outputs, or guiding experimentalists on capturing neurons with characteristic neural profiles (using OMP) for specific tasks. We also present novel findings on the influence of input driven dynamics. Specifically, we directly link how non-autonomous ODEs can be embedded in low-rank RNNs through time-varying basis functions. Thus, we link the RNN connectivity to these target ODEs through dynamics along the input dimension and represent them via temporally evolving basis functions. Empirically, we also validate that identical trajectories can be generated from trajectories of moving fixed points, no fixed points or stationary target ODEs. The presence of dynamics in fixed points suggests a potential failure mode in current methods for interpreting such networks' dynamics Sussillo & Barak (2013); Smith et al. (2021). A potential future direction would be to asses how the conversion of non-autonomous dynamics to autonomous dynamics (which adds an extra dimension) influences insights on underlying brain circuit mechanisms. Lastly, we prove the efficacy of our method by applying it to various neuroscience inspired tasks. We note our method outperforms models of similar size/rank trained with BPTT.

We conclude by discussing some limitations of our work. Our method relies on knowing (or estimating via finite differencing methods) the underlying latent dynamic. Additionally, the complexity of our framework lies in defining or estimating this underlying ODE. While in Sec. 5 we provide instances of embedding non-autonomous ODEs, arguably extending this to other higher order systems could be tricky. This might be especially useful while modeling multi-region brain interactions, where the state of the target ODE depends on another complex dynamical system as well. However, overall we believe the new insights provided by our framework outweighs its limitations and provides an exciting set of future directions.

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
