# OpenReview forum: "A New Look at Low-Rank Recurrent Neural Networks"
_ICLR.cc/2025/Conference — Submitted to ICLR 2025_

### Official Review · Reviewer_Ke7E · 2024-10-31

**Soundness:** 3
**Presentation:** 3
**Contribution:** 2
**Rating:** 6
**Confidence:** 3

**Summary:**

The authors propose a novel perspective by viewing low-rank RNNs as parameterized ordinary differential equations (ODEs) with nonlinear basis functions. Concretely, given a generic RNN $\dot{x} =  -x +J\phi(x) + I u(t)$ and $z = W\phi(x)$ and assuming that $rank(J) = dim(z)$ (i.e. a low-rank RNN for low dim. output $J=MN^T$).  These dynamics can be equivalently represented as a neural ODE with a single hidden layer (of $dim(x)$) and output dynamics $\dot{z} = -z + n^T \phi(Mz_t + Iv_t)$. The authors then consider the problem to fit $\dot{z} = g(z)$ to a *known* output dynamics $g(z)$. This view allows low-rank RNNs to be fitted using generalized linear regression on $n$ given a fixed set of nonlinear basis functions defined through $M$ and inputs $I,v$. The ability to fit certain dynamics is then limited by the expressivity of the basis functions. The authors propose a greedy method to select a suitable basis based on an orthogonal matching pursuit framework (from a predefined set of random basis functions obtained through randomly sampling M,I).

**Strengths:**

The paper is well-written, sound, and easy to follow. The authors provide examples of how this interpretation/method can be used to, e.g., analyze the expressivity of low-rank RNNs and other properties. The proposed method is efficient and easy to apply to given dynamics, and the paper effectively demonstrates the utility of generating new insights on the expressivity of low-rank RNNs.

**Weaknesses:**

- **Training RNNS**: The manuscript claims to "address the issues of RNN training," but the method effectively only "maps" known low dimensional dynamics to high dimensional recurrent neural activity. These are different problems, aren't they? (If I know the low dimensional dynamics, why would I ever use BPTT to embed it into higher dimensional space?). I find the comparison to Backpropagation Through Time (BPTT) somewhat strange. RNNs trained with BPTT effectively address a different problem: learning dynamics from data points. (at least, I assume this is how the RNNs in Fig 5 are trained, as no details are provided). Isn't it expected that they perform worse in your examples because they have to figure out the right dynamics from the data, while your method just fits to the known "true" dynamics?

- **Novelty**: Both the neural ODE interpretation and fitting method using linear regression are not new (e.g., see Beiran et al. 2021, which also uses least squares regression to fit RNNs to known dynamics). While the authors do cite this related literature, these methodological similarities should be more clearly stated in the manuscript.

- **Limitations**: The method does require a known dynamical system to fit i.e. $g(z)$. This is a strong assumption and limits the applicability of the method.  I am unfamiliar with the neuroscience literature, but this is rarely the case, no? The authors should clearly discuss this limitation in more detail (earlier in the manuscript) and add some motivation on relevance (add some more details about the NFE?).

**Questions:**

- Can the authors provide motivation or introduction to the NFE framework?
- Eq. 11 is missing the residual, no?
- Regarding the identified weaknesses, I am particularly concerned about how RNN training is framed in the current manuscript, given that the approach appears to have limited applicability for training and is mainly suited for analysis (see weaknesses).

Overall, I hence tend to reject this paper. Given that I am not an expert on the relevance of this contribution to the neuroscience topic, I will only vote for a weak rejection.

---

> ### Author Response · Authors · 2024-11-21
>
> We thank the reviewer for their time & valuable feedback on our manuscript. We especially appreciate their positive remarks regarding the clarity of our presentation, the efficiency of our method and its significance in providing novel insights on the expressivity of low-rank RNNs.
> ### Weaknesses:
> **1.Training RNNs:** We thank the reviewer for highlighting this, and apologize for not discussing our comparison appropriately. In **Section 6**, both our framework and the networks trained with BPTT are trained from a set of teacher trajectories, which are simulated from the underlying ODE (eg. for binary decision making - they start somewhere on the grid and eventually converge to one of the two fixed points). However, unlike previous sections, our method here can be broken into two main steps.
>
> **a:** We use Euler integration to compute the value of dz/dt along each point of the trajectory. This depicts our target vector for regression.
>
> **b:** Our basis matrix is evaluated at grid points along these trajectories.
>
> Finally, both our method and the networks trained with BPTT must reproduce a set of target trajectories (eg. blue lines in Fig 5-A) from an input pulse train (eg. representing bit value for each channel over the time interval). We thus believe our comparison is equivalent in that both methods are uncovering underlying dynamics (from data points) that solve the task. We’ve **added additional details on training in SI-F.**
>
> **2. Novelty:** We apologize for the lack of discussion regarding similarity to previous methods. Specifc details are provided here (however we’re working on adding a section in our supplement discussing them further) -
>
> **2.1: Neural ODEs:** Previous NEF literature details this equivalence & we’re working on adding this into the supplement.
>
> **2.2: Beiran et al. 2021:** Our work is inspired by the methods presented in this paper. However, they use networks with constant input (text under eqn 3.3) for majority of their analysis. While universal approximation is in the appendix, there is no mention of static or time-varying basis functions (Section 3, 5 of our paper)/specific function classes **(added SI: A.1 & Fig. SI-2)** that can be approximated, which we believe we’re the first to do. Additionally there is no previous work on finding smallest networks (Section 4 of our paper) using this regression. We've **updated Sec 1,7** to discuss these points.
>
> **3. Limitation: known dynamic g(z):** We apologize for the lack of clarity as **our method can be used without explicitly knowing g(z)** (i.e as long as it can be approximated – see 1. Training RNNs). **We’ve updated our discussion section to highlight this**. Additionally, from a neuroscience perspective **interpreting the recurrent connectivity & linking it to the dynamics remains an important question**. Specifically, our work builds on the theoretical foundations laid out in task training RNN literature (e.g. Mante & Sussillo 2013, Sussillo & Barak 2013, Langdon & Engel, biorXiv 2022, Luo, Kim, et al.,biorXiv 2023) & low-rank RNNs (e.g Mastrogiuseppe & Ostojic, 2018; Beiran et al., 2021; Dubreuil et al., 2022; Valente et al., 2022) which focus on training RNNs to perform exactly the tasks in Sec 6 of our paper. Thus while the dynamics are known, our perspective provides novel representational & interpretational insights.
>
> ### Questions:
> **NEF:**  We thank the reviewer for bringing this to our attention. We’re working on including this in our supplement.
>
> **Eqn 11:** Thanks for pointing this out, Eqn 11 is missing terms! We’ve **updated it to reflect the necessary changes.**
>
> **Training v/s analysis:** We apologize for the confusion & hope our discussion above (1. Training RNNs, 3. Limitation:known dynamic g(z)), clarifies the applicability of our method. Additionally, we note alternate training strategies such as FORCE/Full-FORCE, BPTT all require a teacher signal (like our method) & learn either a readout weight/the network connectivity. Our method also provides a way to learn the network connectivity (i.e an appropriate weight vector/matrix n/N), without needing a readout weight. Taken together our method provides an alternate training strategy (inspired by NEF), with the additional advantage of improved interpretability & analysis.
>
> We would humbly ask the reviewer to consider increasing their score if they are persuaded by our clarifications and additional analyses. Thank you again for the helpful comments and suggestions.

---

> > ### Comment · Reviewer_Ke7E · 2024-11-23
> >
> > I thank the authors for their clarifications. This addressed some of my concerns, and the manuscript improved in discussing related work. The explanation of the experimental comparison and how their method is compared to BPTT did clarify this aspect of the work. As clarified by the authors, training from data requires a two-step procedure:
> >
> > - (i) Estimate the vector filed $\hat{g}(z) \approx g(z)$ from data.
> > - (ii) Fit an RNN using the method proposed by the paper.
> >
> > This, however, does only partially address my concern about how the paper frames the "training" aspect, as what is *trained* from data in the above case is the *vector field estimator* $\hat{g}(z)$ and the performance of the RNN is determined by how accurate this estimate is.
> >
> > In a low-dimensional setting, with "noise-free" data points on a high resolution (small dt), the finite difference approach will perform well (as also shown by the experimental setting). However, this approach hardly translates to high-dimensional settings or noisy or low-resolution signals. This approach is further very related to "vector field regression" approaches, usually applied to learn ODEs from data [e.g., 1,2] (which should be mentioned).
> >
> > To my knowledge, the application of this to learn low-rank RNNs is novel and interesting. The manuscript provides a good job of showing how the method can be used given known dynamics (i.e., to fit it to an ODE or an estimate of an ODE). However, as an "alternate training strategy" compared to BPTT, the manuscript only considers one example (which is highly beneficial for the proposed approach). I still think that this requires some calibration, as the training aspect, to some degree, boils down to estimating an ODE with some method and using the proposed method to obtain an RNN.
> >
> > [1] Bhat, Harish S., Majerle Reeves, and Ramin Raziperchikolaei. "Estimating Vector Fields from Noisy Time Series." 2020 54th Asilomar Conference on Signals, Systems, and Computers. IEEE, 2020.
> > [2] Heinonen, Markus, et al. "Learning unknown ODE models with Gaussian processes." International conference on machine learning. PMLR, 2018.

---

> ### Author Response · Authors · 2024-11-25
>
> We thank the reviewer for their detailed and thoughtful response & accurate summarization regarding our training clarification.
>
>   - **1.** The reviewer is correct: our method does depend on how accurate the estimate of the underlying ODE is. It is also true in the case of very noisy/sparse resolution our characterization would have larger errors than those shown in the experiments. However, we have **added an extra simulation Fig SI-10 to show our method can easily be extended to the high-dim noisy data setting.** Specifically, we simulate high-dimensional data from the low-dimensional trajectories and add gaussian noise to this **(details in SI-F.1).** This resembles noisy rate based neural data. To apply our framework to this high-dimensional noisy time-series, we project the data on the top PC. This allows us to recover a noisy estimate of the true underlying trajectory. Our framework can now be applied on these recovered trajectories.
>
>   - **2.** Additionally, we provide *two examples (3 bit flip-flop and bi-stable ODE) comparing our framework to BPTT* – **with the main aim of expressing how our method can be used to train low-rank RNNs for neuroscience tasks** (note in such tasks the dynamic is low-dimensional while the observation, i.e neural recordings could still be high dimensional) **in significantly lower time, while maintaining high interpretability (through the geometric interpretation of basis functions).**
>
> We thank the reviewer for pointing us to “vector field regression” approaches - **we have added this into our discussion in Sec 7 and clarified the added advantage of our framework as an alternative training strategy (with the caveat of knowing/being able to effectively estimate the ODE).**
>
> Finally, we agree with the comments made by the reviewer, but hope our clarifications/reframing expresses the novelty of our work. Additionally we hope our extended simulations showcase the novelty and insights of our work and the ability to adapt our system to multiple settings. We’d kindly request the reviewer to consider adjusting their score if they’re satisfied with our responses.

---

> > ### Comment · Reviewer_Ke7E · 2024-11-28
> >
> > I thank the authors for the review and for addressing the points I raised (at least to some degree).
> >
> > The revised version has strongly improved how the proposed method is presented (specifically in the Abstract/Introduction). I think Sec. 6 could still be improved, e.g.,by explicitly describing how the comparison to BPTT is made (i.e., by stating that in this setting, the vector field to which the RNN is fitted with the proposed method is estimated using finite differences). I appreciate the new results in the noisy high-dimensional setting (which are, however, somewhat limited). These (and also all the other new results) should be referenced in the main manuscript together with the main insight (currently, only Appendix A/B are referenced).
> >
> > Overall, the new version addressed my main concern, and I hence increased my score.

---

> > > ### Author Response · Authors · 2024-11-29
> > >
> > > We deeply thank the reviewer for their positive remarks on the improvements in our paper and for raising their score.
> > > We will certainly add additional clarifying text into Sec 6 and reference all the updated analysis in the final version of our paper as per the reviewers suggestion!
> > >
> > > Once again, we're grateful to the reviewer for their thoughtful response and in helping us improve our manuscript!

---

### Official Review · Reviewer_uMiE · 2024-11-04

**Soundness:** 3
**Presentation:** 2
**Contribution:** 3
**Rating:** 6
**Confidence:** 4

**Summary:**

Rank-$R$ RNNs are universal approximations of arbitrary $R$-dimensional dynamical systems that are (typically) trained with back propagation. The NEF conversely approximates dynamical systems with known equations, using RNNs with low-rank structure, by solving a least-squares minimisation problem. In this contribution low-rank RNNs are fitted to known flow-fields by solving a NEF style optimisation problem, with some focus on obtaining networks with small amount of units.

**Strengths:**

- The work aims to connect some existing lines of work. While I am of the opinion that much of this is was introduced in past work (see weaknesses below), writing out connections explicitly can be useful.
- A new method is introduced for fitting small (in number of units) low-rank RNNs to neural data.
- The paper reads well and the figures do a good job of illustrating the points made.

**Weaknesses:**

My main concern with this paper is that it presents multiple results as new, even though they might not be — sometimes with statements that appear to be incorrect. Given that the paper has “a new look” in the title, I expected more novel insights. I think some rewriting and rephrasing is important, after which this can paper could turn in a useful and nice overview of recent lines of work with some new ideas.

--------

1. Non-linear recurrent dynamics are commonly used in the NEF

>  (Line 073) while the NEF has been widely explored, most studies have focused on feed forward-networks, comparatively few studies focused on non-linear recurrent dynamics

This statement seems false, the NEF is used to embed (non-linear) low-D functions in high-D networks, which can be (and are often) straightforwardly made recurrent, for instance when used for integration / memory, or pattern generation [1]. Most of the online tutorials using the python implementation of NEF (nengo) also deal with time-varying dynamical systems, e.g., non-linear oscillators or the chaotic Lorenz attractor: https://www.nengo.ai/nengo/examples.html.

--------

2. On fitting low-rank RNNs to flow-fields with regression

  - 2.1 The NEF can (and is) also straightforwardly used with rate neurons [1], at which point it becomes at least very close to your framework (it solves the same regression problem). While the NEF crowd typically doesn’t explicitly call their models low-rank RNNs, is there any significant difference with your framework besides the naming?

  - 2.2 In Ref [2] low-rank RNNs were also fit with linear regression to known flow-fields (Section 6). While the parameterisation is slightly different, the general idea is very much the same (and there also the connection to NEF was pointed out).
--------

3. On new insights into universality

Ref [2] also shows that low-rank RNNs with inputs / biases are universal approximations (including for exactly the same system as is used in this paper). It also includes many results relating to symmetry when $\tanh$ is used.

--------

4. On new insight into input driven dynamics

Ref [3] Fig. 1, shows very similar example flow-fields to Fig. 4 in this paper, and used them to make the almost the same point about identifiability. What is exactly the new insight here compared to Ref [3]?

--------

5.  On Low-rank RNNs as neural ODEs

The fact that Eq. 7 is a neural ODE with one hidden layer was explicitly pointed out in previous work (e.g., in the discussion of [4]). The additional insight of the connection to neural ODEs, the literature of which largely focuses on adjoint methods, is in any case also not completely clear to me (some insights by using the adjoint method in relation to low-rank RNNs were obtained in Ref. [5]). I think the main (known) observation to be made here is that Eq. 7 is a universal approximation with one hidden layer.

--------

6. On non-linear oscillators

> ( Line 274) Note that this 2D system is highly nonlinear—unlike recent work focused on oscillations in linear dynamical systems

This statement is misleading. A quick search should give one many studies that investigated oscillations / limit cycles in non-linear RNNs (including low-rank ones!). Ref [2] also derived a limit cycle oscillator in low-rank RNNs.


--------

Refs

[1] Stewart, 2012. A Technical Overview of the Neural Engineering Framework

[2] Beiran et al., 2021. Shaping Dynamics With Multiple Populations in Low-Rank Recurrent Networks

[3] Galgali et al., 2023. Residual dynamics resolves recurrent contributions to neural computation

[4] Pals et al., 2024. Inferring stochastic low-rank recurrent neural networks from neural data

[5] Pellegrino et al., 2023. Low Tensor Rank Learning of Neural Dynamics

**Questions:**

1. Is the introduced method for finding small RNNs guaranteed to converge to the smallest RNN? Or can it converge to a local optimum?

--------

2.
> (Line 277) This target ODE is not radially odd-symmetric, so once again (although not shown here), embedding the system in a low-rank RNN fails if we do not include inputs.

 It is unclear to me how to reconcile the statement with ref [2], where a rank-2 RNN with $\tanh$ units (without biases) was derived such that it implements a similar limit cycle.  One answer could it be that (unlike stated in the text) the system actually is odd-symmetric?  From a quick try, converting your system to back to cartesian coordinates using $z_1 = r\cos(\theta)$, $z_2 = r\sin(\theta)$, I get: $\dot{z}_1 = az_1-z_2$, $\dot{z}_2 = az_2+z_1$, with $a=\frac{1-r^2}{r}$, which satisfies $F(-z_1,-z_2) = - F(z_1,z_2)$.

---

> ### Author Response · Authors · 2024-11-22
>
> We thank the reviewer for their valuable feedback, suggestions, & positive comments on the clarity of our presentation & novel method for identifying the smallest RNN. We were unaware of some past findings the reviewer highlighted & appreciate the opportunity to refine our claims & appropriately credit previous work.
> ### Weaknesses:
> **1. Non-linear recurrent dynamics in NEF:** The reviewer is correct & we apologize for the confusion. [1] shows applications of non-linear recurrent dynamics. We didn’t intend to claim NEF is limited to feed-forward networks, only that most applications we’d seen fell into that category.  However, we realize this was inaccurate & we’ve **updated Sec 1,7 to correct this.**
>
> **2. Fitting low-rank RNNs with regression:**
>
>    **2.1 NEF:** The reviewer is correct: NEF can & has been used with rate neurons. Part of our motivation was the (what seemed to us) relative disconnect between low-rank RNN work & the NEF work. As the reviewer noted, networks constructed by NEF are low-rank, but haven’t (to our knowledge) been framed as low-rank RNNs. Similarly, we hadn’t seen previous low-rank RNN work making the explicit connection to NEF (we’re grateful to pointing us to [2])! **Overall, we feel this connection isn’t as well known as it should be, evidenced by most papers on low-rank RNNs using BPTT & feel our work provides intuition for the role that individual neurons play in low-rank RNNs (e.g. Fig 1 our paper).** We apologize for the oversight & will correct our text to focus more on novel insights (e.g. OMP).
>
>    **2.2 [2]:** Our work is inspired by [2], however, they use networks with constant input (text under eqn 3.3) for majority of their analyses. While universal approximation is in the appendix, there is no mention of basis functions/specific function classes (**added SI: A.1 & Fig. SI-2**) that can be approximated, which we believe we’re the first to do. Also, there is no previous work on finding smallest networks (OMP) using this regression.
>
> **3.New insights on universality:** We thank the reviewer for raising this point
>    - **1.** As discussed above, [2] considers constant inputs for majority of their analyses. Second, their exploitation of symmetry arises exactly because of this. Eg: [2] Section 5.1: “if a network generates one nontrivial stable fixed point, additional stable fixed points appear at symmetric points in the collective space”. Our results explicitly clarify why this occurs: namely, when the RNN has an odd-symmetric nonlinearity in the absence of inputs. (However, it would not generally hold for other nonlinearities).
>
>    - **2.** [2]: no mention of basis functions, which we feel provides nice linear algebraic/geometric insight into how a low-rank RNN constructs a representation of an ODE. Our insight is in: viewing neuron specific inputs as shifted basis functions which is arguably more intuitive. Additionally, we expand on how specific non-linearities ([2] restricts analysis to tanh - text under eqn 2.1) can approximate specific function classes better, which to our knowledge hasn’t been previously reported (**added SI: A.1 & Fig. SI-2**).
>
>  - **3.** Previous work doesn't explore how universal approximation can be met with a sparse/finite basis set, which we show through OMP (within a margin of error).
>
> 4. **New insights input driven dynamics:** We apologize for the confusion. Our goal was to validate our results grounded in accepted theories (eg, identifiability in [3]). **Our novel insight is in interpreting time-varying dynamics in low-rank RNNs through time-varying basis functions (i.e contribution of each neuron’s response): top row Fig 4.** Also, prior work ([3], or others) to our knowledge, doesn’t provide explicit ODEs for such “dynamic attractors” via moving fixed points/no fixed points, which we address - Eqn 13.
>
> 5. **Neural ODE equivalence:** We thank the reviewer for pointing this out. We weren’t aware of [4] which connects low-rank RNNs and neural ODEs ([2] omits inputs in Eqn 2 & discusses it only in the appendix). We’ve **updated Sec 3 to cite [4], [5]** & clarify the connection. We acknowledge there exist other useful links to the neural ODE literature that we haven’t explored in this work.
>
> 6. **Non-linear oscillators:** The reviewer is correct, we deeply apologize. We meant “nonlinear & non-symmetric” (updated Sec 3.1).
>
> ### Questions:
> 1. **Convergence of OMP:** Thanks for this question. OMP converges to a local optimum.  However, we show performance is highly consistent across runs in simulation **(added SI-E, Fig SI-9).**
> 2. **Line 277; Ref [2]; radially odd symmetric:** We thank the reviewer for pointing this out. The governing ODE for our limit cycle is non-symmetric: \( dz_1/dt = az_1 - z_2 - 0.35 \) & \( dz_2/dt = az_2 + z_1 + 0.5 \), \( a = 1 - r^2r \). Since \( 0.35 \) & \( 0.5 \) don’t depend on \( z_1/z_2 \), the function is **not odd symmetric**. We’ve **added Section A.2 & Fig. SI-3 & updated Section 3.1** to address this.

---

> ### Comment · Reviewer_uMiE · 2024-11-23
>
> **1. regarding the NEF**
>
>  I appreciate the author’s restating in the introduction. However some claims with respect to the NEF are still overstated, e.g.,
>
> > This presents an extension to the NEF method
>
> > Our novelty lies in characterizing a randomized basis for encoding/embedding, thus deviating from the classical NEF setting which carefully crafts how neurons are expected to respond to stimuli
>
> The NEF/nengo typically also starts by sampling random tuning curves / basis encoding function and then solving for the linear decoding functions with least squares (exactly as you do).
>
> I think it is quite crucial to do at least a high-level explanation of the NEF in the main text, and reformulate said sentences in order to provide the reader clarity about what is new (and what is not).
>
>
>
> **2. Fitting low-rank RNNs**
>
> >In this work, we address the issues of RNN training
>
> You only address fitting RNNs to known flow-fields, for which methods without e.g., the problem of exploding gradients do exist [1]. The closest is simple teacher forcing (TF): given a (discretised) RNN/neural ODE $F$ and pairs of $z_t$ and $z_{t+1}$, minimise $\lvert z_{t+1} -F(z_t) \rvert$. This is very similar to what you end up doing (especially when using the finite difference method to estimate $\dot{z}$ from trajectories). The main difference would be that you use the NEF approach of sampling random encoding weights and solving for the decoding weights with regression, instead of finding all parameters with back propagation. It could be good to name TF (and extensions, e.g., [1] also fit one hidden layer ODEs/RNNs) - TF probably would have been a fairer comparison than BPTT, but that likely is out of scope now.
>
> **3. On universality**
>
>  It could be true that no-one named NEF networks low-rank RNNs explicitly, although I am sure people were aware of the low-rank structure of NEF networks (see also e.g., [2]). I do now agree with the authors that the geometrical basis functions explanation / NEF view adds some clarification to [3].
>
>
> **4. Input driven dynamics**
>
> Thanks for clarifying the additional insights to previous work, I now agree there is something new here. I do think the insight that small perturbations lead to different trajectories in different models is similar to [4] (where perturbations from the mean, were used to distinguish input versus autonomous dynamics).
>
>
> **5. Neural ODE**
> Thanks, this is fine now!
>
>
> **6/Q2. Odd-symmetry of the non-linear oscillator**
>
> It is indeed correct that by adding the offset to $z_1$ and $z_2$, the oscillator becomes non-odd-symmetric. For reference, I do want to point out that the new equations are different from the ones on the original manuscript, which were $\dot{\theta} = 1, \dot{r} = 1-r^2$ - which did imply the odd-symmetric system.
>
> > Note that this 2D system is highly nonlinear and non-symmetric — unlike recent work focused on oscillations in nonlinear dynamical systems
>
>  I still think this should be rephrased, you are not the first to get RNNs to display limit cycles (even when adding some offset), maybe it is best to omit the unlike… part altogether?
>
>
>
> **Q1. Smallest network**
>
> The fact that you converge to a local minimum should be named explicitly in the main text in my opinion - the method finds a small network, not necessarily the smallest network
>
> ————
>
> To avoid being overly negative, I do want to highlight that there is definitive value in this work, e.g., the OMP algorithm and the clarifications on universal approximations in low-rank RNNs. The writing is easy to follow and the figures are also great. The current narratives of “we extend the NEF”, and “we overcome limitations of BPTT”, are in my opinion (as I hopefully clarified above), not the ideal way to present these contributions.
>
> ————
>
> [1] Hess et al., 2023. Generalized Teacher Forcing for Learning Chaotic Dynamics
>
> [2] Barak and Romani, 2020. Mapping low-dimensional dynamics to high-dimensional neural activity: A derivation of the ring model from the neural engineering framework
>
> [3] Beiran et al., 2021. Shaping Dynamics With Multiple Populations in Low-Rank Recurrent Networks
>
> [4] Galgali et al., 2023. Residual dynamics resolves recurrent contributions to neural computation

---

> > ### Author Response · Authors · 2024-11-25
> >
> > We thank the reviewer once again for the numerous comments & pointers to previous literature. We’ve **revised the narrative of our paper per the reviewers suggestion, to emphasize our viewpoint provides geometric intuition & have focussed our novelty claims more narrowly on the extensions relating to use of OMP.**
> >
> > **1. NEF:** We deeply apologize for the oversight & have **revised our formulation in Sec 1,7, clarifying that we’re applying the NEF framework to low-rank RNNs, while providing a high-level explanation.**
> >
> > **2. Fitting Low-Rank RNNs:** We thank the reviewer for pointing this out and we’ve now mentioned TF methods in our manuscript! However, we do differ from [1] significantly (and other TF like methods eg FORCE/Full-FORCE) -
> >    - **1.** [1] restricts their analysis to a specific class of piecewise linear RNNs (PLRNNS) since they’re mathematically tractable (text above Eqn 10 [1]) and have highest performance (last para of discussion) on them. Our framework is easily expanded to other non-linearities, with clarifications on performance changes across such non-linearities.
> > - **2.** Interpretation of the low-d structure isn’t obviously clear from these networks (i.e some post-processing is needed). Our NEF inspired framework allows interpretability, while still overcoming the issues of training achieved by TF.
> > - **3.** TF presented in [1] is similar to the FORCE algorithm discussed in our paper ( Sec 1, SI-C.1) in that they require multiple epochs of training, and are sensitive to hyper-parameters (alpha in [1], ‘g’ in FORCE). Our method allows for a single step, closed-form solution.
> >
> > Taken together, **in the absence of the parametric form of the ODE, our method overcomes issues of BPTT in a manner resembling TF methods, with the added advantage of interpretability & reduced computational cost.** We hope our Sec 1, which discusses other training methods along with the interpretability advantages of NEF, presents a clearer motivation for our work. **Per the reviewers suggestion we’ve included a mention to TF methods in Sec 7**.
> >
> > **3. Universality:** We thank the reviewer for acknowledging our work adds clarification to [3]. Also while [2] is low-rank it is never mentioned, which is exactly what our work aims to explicitly do. Furthermore, [2]  focuses on a linear RNN, with nonlinear cases handled only in the appendix.
> >
> > **4. Inputs:** Thank you, we’ve **added a citation to [4] in Sec 5 (last line).**
> >
> > **6: Odd symmetry nonlinear oscillator:** The reviewer is correct, our original equations were misleading, we’ve **added the correct equations in SI-A.2 & updated Sec 3** based on the reviewers suggestion.
> >
> > **Q1 Smallest Network:** The reviewer is correct and **we’ve added text clarifying our result (text above Algorithm 1 in Sec 4).** Specifically, given a specific basis set (on which OMP is run) it will find the smallest network. However, if this original basis set is changed -- it might find a smaller/larger network (however this would still be the smallest network within this new basis set). As shown (SI-E, Fig SI-9), if the basis set is equivalent (in terms of range it covers), the algorithm is consistent across runs (i.e the exact basis might be different but it achieves similar loss with the same number of neurons). We thus believe claiming the smallest network is appropriate, with the added explanation of it being the smallest network within the basis set from which it is computed.
> >
> > We sincerely appreciate the reviewers overall positive comments on the value of our work and presentation clarity and are grateful for their feedback in helping us improve the quality and presentation of our work. We hope the revised changes reflect this and would humbly request the reviewer to consider adjusting their score if they’re satisfied with our reframing!

---

> > > ### Comment · Reviewer_uMiE · 2024-11-26
> > >
> > > The paper has improved, and I have slightly raised my score. However, while the authors have addressed individual sentences in the text, the paper would in my opinion still be much improved if the **general story line more clearly focus on the new contributions** (such as geometrically interpreting low-rank RNNs through basis functions, OMP) – which in my opinion by themselves would make for a solid paper.
> > >
> > > For instance “proposing an alternate RNN training strategy” that overcomes issues of RNN training is not the main contribution, but is one that is still very prominently highlighted throughout the introduction, e.g.,
> > > > In this work, we propose an alternate RNN training strategy that is deterministic, requires low
> > > compute time while also addressing interpretability issues of such models
> > >
> > > First, I don’t think any of the RNNs in the paper could not have been trained with simple teacher forcing, which in its simplest form minimizes exactly the same loss as is minimized by Eq. 9 (i.e., when using generalised teacher forcing with alpha = 1 of [1]). It is true that this would take multiple iterations when using gradients instead of regression – however these iterations will be cheap, as there is no back-prop *through time* and we can parallelize over observed $z$’s, similar to your method. Second, the training method (except for the OMP variant) is largely not something “new” you propose.
> > >
> > >
> > > In you last comment you state that:
> > > >Also while [2] is low-rank it is never mentioned, which is exactly what our work aims to explicitly do. Furthermore, [2] focuses on a linear RNN, with nonlinear cases handled only in the appendix.
> > >
> > > Here I’m quoting from the discussion of [2]:
> > >
> > > > The NEF low-rank connectivity can be interpreted as linearly mapping between low- and high- dimensional dynamical systems, with the surprising result that this transformation also works for nonlinear neural dynamics.
> > >
> > > ---
> > >
> > > (btw the reference list is gone in the uploaded pdf)

---

> > > > ### Author Response · Authors · 2024-11-26
> > > >
> > > > We sincerely thank the reviewer for their detailed response consideration. We have updated some of our text (abstract, introduction and discussion) to highlight OMP and the geometric interpretation as suggested by the reviewer.
> > > >
> > > > 1. proposing an alternate RNN training strategy: We've updated our discussions to reflect while we present a closed-form solution, however that isn't the main contribution of our work as suggested by the reviewer. Specifically we include text stating this training scheme alleviates issues of back-prop (like many other existing methods), but provides a more intuitive portrait on how dynamics are embedded via basis functions.
> > > >
> > > > 2. Discussion on [2]: Thank you for pointing this out, we have revised the text in our paper that suggests this!
> > > >
> > > > 3. Reference list: Thank you! we've updated our submission to include it!
> > > >
> > > > Once again we deeply thank the reviewer for their detailed comments and sincerely appreciate their effort in helping us improve the quality of our presentation and submission.

---

> > > > > ### Comment · Reviewer_uMiE · 2024-11-27
> > > > >
> > > > > I again thank the authors for their willingness to take into account the reviewer's suggestions.
> > > > >
> > > > > Reviewer eZS4 comment clearly summarises my current thoughts:
> > > > > > Overall, I still believe the feedback was incorporated into the manuscript at the sentence level rather than holistically.
> > > > >
> > > > > I also do strongly think this could be a better paper with more holistic rewriting, and e.g., immediately focuses on interpreting and implementing known dynamics (instead of task-training), and includes some explanation of the NEF in the main text. Also largely the "example" sentences that specifically where pointed out by reviewers are changed, there are still statements like:
> > > > > > Our framework can thus be used as an alternative to task-training in neuroscience, to model novel behavioral tasks.
> > > > >
> > > > > This (a; as other reviewer's have also pointed out) is only true if the desired underlying dynamics are known (which generally isn't the case), and b) is also provided by TF.
> > > > >
> > > > > Overall I am leaning slightly towards rejection, but I do think the paper has improved and contains no large inaccuracies - if other reviewers vouch for acceptance I would concur.

---

> > > > > > ### Author Response · Authors · 2024-11-28
> > > > > >
> > > > > > We sincerely thank the reviewer for their positive comments and willingness to help us improve our work.
> > > > > >
> > > > > > Holistic changes: We've now revised our text to improve the overall flow of our manuscript as per the reviewer's suggestions. Specifically, we've included a more detailed explanation of NEF in the main text, and rephrased our contributions in our discussion significantly based on the reviewers suggestions and comments.
> > > > > >
> > > > > > Additionally, we understand the reviewer's suggestion on removing task-training from our introduction, however we believe this is a critical component of the literature upon which our paper builds. Specifically, most of the low-rank RNN theory focusses on the tasks detailed in these papers (and similarly so do the ODE examples described in our paper). Additionally, since such networks can be trained in a variety of ways, with majority of the recent literature (specifically for task-trained RNNs) focussing on BPTT, we believe this adds significant context into why we deviate from BPTT (and how our framework, like some of the other methods has advantages over it in certain scenarios). We however agree this training scheme isn't the main contribution of our work and hope our rephrasing clearly highlights this as well!
> > > > > >
> > > > > > alternative to task-training: The reviewer is correct, we've reframed our contributions in our discussion to correctly highlight the focus of our work and the interpretational insights that are novel.
> > > > > >
> > > > > > We deeply thank the reviewer for their suggestions and recommendation of our paper. We hope this revision of our manuscript indicates a holistic change based on all the suggestions and comments!

---

> > > > > > > ### Comment · Reviewer_uMiE · 2024-11-28
> > > > > > >
> > > > > > > Thanks again! I now also recommend acceptance.

---

> > > > > > > > ### Author Response · Authors · 2024-11-29
> > > > > > > >
> > > > > > > > Thank you, we sincerely appreciate the reviewer recommending our paper for acceptance!

---

### Official Review · Reviewer_eZS4 · 2024-11-04

**Soundness:** 2
**Presentation:** 2
**Contribution:** 2
**Rating:** 6
**Confidence:** 3

**Summary:**

The authors make use of Neural Engineering Framework to present a different perspective on how to embed a given dynamical system into a low-rank RNN. They put emphasis on neuron-specific inputs or bias to make the low-rank RNNs universal approximators. The paper claims concerns on the fixed-point analyse framework of RNNs which is one of the mainstream type of reverse-engineering method in computational neuroscience.

**Strengths:**

The provided perspective of latent space view is beautiful and shows connections between neural ODEs and low-rank RNNs.

The paper demonstrates clear examples of indistinguishable dynamical systems.

Proposed learning algorithm is fast and efficient.

**Weaknesses:**

First, the problem that is concerned is to embed a _given_ dynamical system to a low-rank RNN. However, this approach, as also acknowledged by the authors, limits the applications. I believe this restriction is big and should be discussed by authors more. Currently, I do not think it is addressed enough since it is one of the *main* issues of the paper. I think it would be better to discuss this earlier. It is clear that this method cannot be used to find the dynamics required for a novel task --which is arguably, one of the most important use-cases of RNNs. So it can be only used for the tasks that we already know (or at least have hypotheses) its dynamical system. I would expect from authors to have more (and more importantly, distinct) use-cases of their algorithm for science purposes.

Second, I do not believe comparing your algorithm with BPTT is fair, because you do not provide what BPTT provides (see above). Plus, I couldn't see the details of how you train the two tasks using BPTT. Is there a clear way of embedding a given dynamical system into the RNN using BPTT? If so, I think if you do this to compare your algorithm with BPTT, it would be better. If not, I think this can be put into supplementary results but I don't think it is important.

Third, you state "A low-rank RNN is therefore not a universal approximator unless it has inputs, or equivalently, different biases or offsets to each neuron." I think this is an odd point to make. Because,  1-) An RNN is only a universal approximator when it has inputs or different biases. 2-) A low-rank RNN is strictly a subset of RNN. And so therefore, if an RNN does not have a property under some conditions, and so low-rank RNNs are. So I do not understand the point made by authors. You emphasize this point in your abstract as well, saying, "our perspective clarifies limits on the expressivity of low-rank RNNs", which I think is misleading, because it is not a problem of low-rank RNNs but RNNs in general.

Fourth, I liked the Figure 4, but I don't think this analyses could only done using your perspective. I think the field already knows the point you make, but you make your point nicely so it is still important. So you say "Our discussion on non-autonomous dynamics suggests, how similar trajectories can be observed in the presence of dynamical fixed points/ no fixed points at all." but does this analysis allowed by your algorithm or is this a distinct point made in your paper besides the provided algorithm? Moreover, you can transform non-autonomous RNN into an autonomous one. This is a very important point which should introduce philosophical questions on your point from the neuroscience point of view.

I think before introducing the Eq 9, you can talk (very briefly) about least-squares regression.

You put an introduction to low-rank RNNs but not NEF. I think the paper would benefit a lot if you also put emphasis on NEF more (mathematically) to make your connection more clear.

Typos (did not affect my score):
Line 097, ReLu -> ReLU
Line 242 phi(x)
Line 715, "we thus use present a general"
Line 777, "(A) Bi-Stable Attractor ODE and (B) Line Attractor ODE." unusual capital usage
Line 796, M is m1, m2, m2, and better to put space between them.
Line 801, "Following our discussion o the"
Line 812, "Data Simulated from trajectories", didn't understand the choice of capitals
When you write Lorenz, you should use capital L.

**Questions:**

What would be a way of finding the smallest RNN using BPTT? Does your algorithm makes it faster, or it provides a theoretical way of doing it as well, can you put more discussions on this?

You wrote an algorithm section for finding the smallest RNN, can you also do it for your main algorithm as well? I think this would clarify a lot.

Under which conditions low-rank RNNs are equivalent to neural-ODEs? If the latent space has noise, are they still equivalent? I would expect a more rigorous way of stating this.

You say "(although not shown here) ... fails", I wondered how does it look like when it fails.

---

> ### Author Response · Authors · 2024-11-21
>
> We thank the reviewer for their careful consideration & insightful feedback. We appreciate their positive remarks on the beauty & efficiency of our perspective, & clarity of our examples. Below, we address the reviewer's concerns & suggestions.
> ### Weaknesses:
> **Embed dynamical system in low-rank RNN**: The reviewer is correct: our method can be used for tasks with known dynamics. However, **in Section 6, our framework & networks trained with BPTT infer dynamics from the same set of training trajectories**. The dynamic (regression target) is approximated using finite differencing (euler method). **Thus even when the ODE is unknown, its value is estimated at points along the trajectory. We’ve added section SI-F to clarify this.**
>
>    a. **Use-cases:** Task-trained RNNs interpret governing dynamics of specific tasks linked to brain computation (e.g., Kanitscheider & Fiete 2017; Turner, E, et al. 2021; Luo, Kim, et al. 2023). However, degeneracies in the network/task solution space often lead to incorrect conclusions. We propose designing ODEs to solve tasks, which our method embeds into the RNN. We’ve **added two additional dynamical portraits for evidence accumulation tasks (Fig SI-5)**.
>
> **BPTT Comparison:** Our framework & BPTT networks are trained from a set of teacher trajectories simulated from an underlying ODE. (eg.: decision making, trajectories start from a random initial location on \z\ until convergence to a fixed point). Unlike previous sections, we used finite-differencing to compute \( dz/dt \) from sample trajectories. We then fit the \(n\) weights by regressing these derivatives against a design matrix formed by evaluating the nonlinear basis functions at the points in these trajectories. Thus, both training methods (ours & BPTT training) had access to the same training data.
> Both our method & BPTT-trained networks reproduce target trajectories (e.g., blue lines in Fig 5-A) from an input pulse train (e.g., representing bit values). Thus, both methods uncover underlying dynamics that solve the task. We’ve **added details in SI-F to clarify this.**
>
> **Universal approximator:** Reviewer is correct: low-rank RNNs are a subset of RNNs. However, universal approximation is largely overlooked in the low-rank RNN literature. For eg., Beiran et al., 2021 use networks with constant input (text under eqn 3.3). While universal approximation is mentioned in the appendix, there is no mention of basis functions/specific function classes that can be approximated, which we’re the first to do. To further clarify this **we’ve added SI: A.1 & Fig. SI-2.**
>
> **Fig 4 & contribution:** Reviewer is correct: dynamics in Fig 4 can be found via other methods. To clarify – our goal was to validate our results grounded in field-accepted theories (eg. need for perturbations). However, **unlike other methods, our framework adds novel interpretations of time-varying dynamics through time-varying basis functions. (i.e contribution of each neuron’s time-varying response in approximating the dynamic).**
>
> **Non-autonomous to autonomous:** Doing this adds an extra dimension to the system which could play a critical role in neuroscientific insight. While not explored here, we’ve **added this in our discussion section.**
>
> **NEF & Regression:**  Thanks for this suggestion, we’re working on adding this in the supplement.
>
> **Typos:** Thanks for pointing these out! We’ve **corrected them in the revised paper.**
>
> ### Questions:
> **Smallest RNN using BPTT & comparison**: Current methods train networks (different ranks & sizes) with BPTT & pick those with best performance. Dynamics solving tasks are then interpreted via dim reduction (eg. Valente et al. -NeurIPS '22, Sussillo, Kanaka, et al. Neuron 2016,Sussillo, D., & Abbott, L. F., Neuron 2009). We instead directly embed hypothesized ODEs governing tasks (dim=rank) without training multiple networks. We use regression to learn the required weighting & provide theoretical & empirical results (OMP), identifying the smallest RNN approximating a function. We’ve **added tables comparing training times for networks (trained with BPTT) in SI-F, Table 1,2.**
>
> **Algo for main method & OMP:**  Our primary method (Sec 3 & 3.1) involves:
> 1. Generating a random basis matrix.
> 2. Using regression to learn the weight vector/matrix.
>
> Alternatively, Sec 4 provides an algorithm to optimally create a sparse basis matrix, .
>
> **Neural ODE equivalence:**  Previous NEF literature details this equivalence & we’re working on adding this into the supplement. In case of noise in the latent space, our framework would map to neural SDEs.
>
> **“(although not shown here) ... fails”:** We’ve **added a section (A.2, Fig SI-3)** on why inputs are needed in the 2D case when the function lacks odd symmetry.
>
> We humbly ask the reviewer to consider raising their score if they feel our revised paper (with added simulations, analyses, clarifications, & discussions) would make a worthwhile contribution to the meeting.

---

> ### Comment · Reviewer_eZS4 · 2024-11-25
>
> I sincerely thank the authors for their efforts to improve their paper with the suggestions from all reviewers.
>
> BPTT Comparison: Thank you for clarifying this point. I think it would be important to add a sentence in the main paper under the BPTT section, such as "trained from a set of teacher trajectories simulated from an underlying ODE." Including this directly in the main text, rather than explaining it solely in the supplementary material, would improve clarity for readers.
>
> Universal Approximator: I understand and appreciate the authors' contributions, which are clearly valuable, but some of the wording might lead to confusion. For example, the statement "More generally, our perspective clarifies limits on the expressivity of low-rank RNNs, such as the fact that without inputs, a low-rank RNN with sigmoidal nonlinearity can only implement odd-symmetric functions," might make it seem like this limitation is unique to low-rank RNNs.
>
> It could be true that universality hasn’t been discussed in the low-rank RNN literature before, but as we’ve discussed earlier, low-rank RNNs inherit some of the same challenges as general RNNs. In this case, the fact that low-rank RNNs are not universal approximators without inputs or biases also seems to be an issue for RNNs in general. So, when the authors write "limits on the expressivity of low-rank RNNs," it could come across as misleading, as it suggests this is a problem unique to low-rank RNNs. I hope I’m not misunderstanding here. I think it would be better to focus more on the geometric interpretation, which is the authors' main contribution. Alternatively, if the authors want to emphasize this point, I suggest clarifying it in the abstract as well, as a misleading abstract would not be helpful for the paper.
>
> Overall, the main issue seems to be how the contribution is framed. For example, the authors provide a formal connection between neural ODEs and low-rank RNNs. However, the wording, "... and shows that a low-rank RNN corresponds to a neural ODE with a single hidden layer [REFS]," could be revised to better reflect their work. It might be clearer to say something like, "This connection has been noted in previous work [REFS], and here we formalize it." This is just one example, but as other reviewers have also pointed out—and I agree—the contributions and the writing don’t fully align.

---

> > ### Author Response · Authors · 2024-11-26
> >
> > We deeply thank the reviewer for their thoughtful response and suggestions. We have since **updated our abstract, introduction and discussion, as well as text on universality to better reflect the reviewer's concerns.**
> >
> >  BPTT Comparison: Thank you! We have added text in the Sec 6 to specifically state this.
> >
> >  Universal Approximator: We appreciate the reviewer's suggestions and have updated our comments on this to better reflect the need for inputs isn't restrictive to low-rank RNNs, while that is the specific class of RNNs that our paper focusses on (given the neuroscientific appeal for such models).
> >
> > Once again we thank the reviewer for their time and thoughtful evaluation of our paper and hope our revisions address some of the writing concerns presented.

---

> ### Comment · Reviewer_eZS4 · 2024-11-27
>
> I appreciate the author's engagement during the rebuttal process.
>
> Universal Approximator: I believe this change has clarified the paper's contributions. If you feel it is sufficient to include this information in the paper but not in the abstract, please feel free to disregard my earlier suggestion about including it in the abstract. As it stands, the abstract with the parenthesis reads awkwardly unless rewritten for better flow.
>
> Overall, I still believe the feedback was incorporated into the manuscript at the sentence level rather than holistically. The other example is, the sentence, "Additionally, although not explored here, the conversion of non-autonomous dynamics to autonomous dynamics could play a critical role in neuroscientific insight," disrupts the flow and seems to have been added later. I kindly suggest that you feel free to ignore my recommendation if you disagree—I am entirely fine with that if you provide your reasoning. Otherwise, I encourage you to consider incorporating the feedback more holistically.
>
> The figures are great, explanatory, and demonstrate the contributions. I increased my score slightly; however, I still think **the paper is not yet in its best form due to a reduced flow caused by sentence-based additions and would benefit a lot if the structural writing changes were made.**
>
> Additional Note: Still have some typos: Line 200, ReLu, Line 242 phi(x).

---

> > ### Author Response · Authors · 2024-11-28
> >
> > We thank the reviewer for their positive remarks and for adjusting their score. We've updated our text to improve the overall flow of our paper as per the reviewer's suggestion.
> >
> > Universal approximator: Thank you, we've cleaned out these sentences to reflect our contribution and not provide any misleading arguments (by specifically incorporating text in the main paper).
> >
> > Non-autonomous to autonomous: The reviewer was correct in their assessment, however as suggested in our text this is an interpretational insight that we haven't explored in this version of the paper. We have however updated the text to improve the overall flow of the paper.
> >
> > Typos: We've corrected them in this revision, thank you so much for pointing them out!
> >
> > Once again, we're extremely grateful to the reviewer for their detailed comments and feedback and hope this revision incorporates the suggested structural changes in our writing more holistically (we've added additional NEF related details and clearly highlighted our contributions in the discussion section).

---

### Official Review · Reviewer_RBkH · 2024-11-05

**Soundness:** 2
**Presentation:** 2
**Contribution:** 3
**Rating:** 3
**Confidence:** 4

**Summary:**

This paper advances the understanding of low-rank RNNs by introducing a new theoretical framework, an efficient training approach, and empirical validation. These contributions improve both the interpretability and representational capacity of low-rank RNNs, demonstrating their ability to embed complex nonlinear dynamical systems through a randomized basis. The authors emphasize the importance of inputs in capturing odd-symmetric functions, expanding on prior research regarding the universal function approximation properties of these models. Also, their method enables closed-form parameter learning through regression, which greatly reduces the number of parameters compared to traditional methods. Altogether, the paper positions low-rank RNNs as an efficient and interpretable alternative to conventional training techniques.

**Strengths:**

**Originality**: The paper introduces a new perspective on low-rank RNNs by modeling them as low-dimensional ODEs using nonlinear basis functions. This approach links low-rank RNNs to the NEF, which has mostly been used for feed-forward networks. The authors present a more interpretable and efficient training method, avoiding the limitations of traditional gradient-based training.

**Significance**: The significance of the paper lies, particularly, in its potential to address the challenges of training and interpreting RNNs.

**Weaknesses:**

1) The proposed method is not clear enough. For instance:

- **Rank r**. It is unclear how one can systematically determine the optimal rank r for low-rank RNNs in general. It might be easier achievable for benchmark systems with specific trajectories or dynamics, but it is not clear enough for more complicated cases or real-world datasets with higher dimensions. Could you provide guidelines or heuristics for selecting **the rank r** for different types of problems or datasets?

- **Role of different activation functions**. Given the impact of different activation functions ϕ on network performance, what criteria should be used to select the most appropriate activation function for a given task? Are there specific tasks where certain functions consistently outperform others? Could the authors include a comparative analysis of different activation functions on a set of benchmark tasks, showing how performance varies across functions?

- **Probability distribution of the parameters**. Choosing an appropriate probability distribution for the parameters that define the random basis is crucial, as it may directly impact the model's expressivity, learning efficiency, and the stability/convergence of the training process.  Could the authors conduct an ablation study comparing different probability distributions for parameter sampling? Or, is there a data-driven sampling scheme to generate the random basis (i.e., a method for adapting the sampling distribution based on the characteristics of the dataset or task, e.g., similar to https://arxiv.org/abs/2410.23467)?

2) The paper claims that the proposed method converges faster than traditional BPTT methods, but it lacks a detailed analysis of computational efficiency, including **training time**. Could the authors provide a more detailed analysis of the actual training times across different models and tasks?

3) Regarding the Lorenz attractor, in *Figure SI-3*, the plots do not allow for a clear comparison between the true and fitted (reconstructed test) trajectories. Could you include additional plots for comparing the true and fitted **chaotic trajectories**, as well as the associated **time series** of the x, y, and z components?

4) The paper mentions the **interpretability** of low-rank RNNs but does not delve deeply into it. Could the authors provide more explanations about the interpretability of their method compared to other methods?

**Questions:**

What additional experiments could be conducted to further validate these findings, particularly in real-world applications?

Please see also "Weaknesses".

**Details Of Ethics Concerns:**

No ethics concerns.

---

> ### Author Response · Authors · 2024-11-21
>
> We sincerely thank the reviewer for their valuable feedback & positive remarks on the originality & significance of our work in advancing efficient interpretation & training of low-rank RNNs. To address the reviewer’s comments, we’ve **conducted new analyses**, detailed below & believe the revised paper is much stronger. We humbly request the reviewer to consider adjusting their score if our response satisfactorily addresses their concerns.
> ### Weaknesses:
> 1. **Method not clear enough**
>
>    a. **Rank \( r \):** We thank the reviewer for calling attention to this.
>
> **a.1. Benchmark systems:** As correctly identified by the reviewer, our approach introduces a new perspective on low-rank RNNs by modeling them as low-dimensional ODEs. Thus given the underlying ODE, the rank of the network equals the dimensionality of the ODE (e.g., Section 3,3.1: 1D, 2D ODE & SI Fig. 3: 3-dim Lorenz system).
>
> **a.2. Real world datasets:** Our work builds on theoretical foundations laid out in low-rank RNN literature (e.g., Mastrogiuseppe & Ostojic, 2018; Beiran et al., 2021; Dubreuil et al., 2022; Valente et al., 2022) & the “task training” RNN literature (e.g., Mante & Sussillo 2013; Sussillo & Barak 2013; Luo, Kim, et al., bioRxiv 2023), which focus on training RNNs to perform the decision-making tasks we consider in Section 6 of our paper. Thus, while these tasks aren’t high-dimensional, they're challenging for animals to perfect & perform. Additionally, we acknowledge the reviewer's criticism on not providing a method to determine the optimal rank. We agree this is an interesting problem, which we don’t yet know how to solve. As far as we’re aware, none of the previous literature provides a theoretical solution to this problem (e.g., current methods try models with different ranks & pick those with the highest performance & interpretability). The significance of our method is: given the ODE of some dimension, we directly embed it into the low-rank RNN (where rank = ODE dim) without needing to train multiple models using FORCE/backprop.
>
>    b. **Activation function role:** The reviewer is correct: different activation functions do influence performance. We've **added a comparative analysis (SI: A.1 & Fig. SI-2)** for two different ODEs, one which is better approximated with Tanh & the other with ReLU. Our analysis suggests the following criteria:
>       - Piecewise linear dynamics can be better approximated by smaller ReLU networks (Similar performance is achieved with larger Tanh networks).
>       - Tasks with more non-linear dynamics are better approximated by smaller Tanh networks.
>
>    c. **Probability distributions:** We’ve **added an ablation study (SI: E & Fig. SI-9)** showing similar performance (similar # of neurons with OMP) on the bi-stable ODE, regardless of the distribution (uniform/standard normal) of the basis functions. Thus, our results suggest: **1.** the specific distribution of basis functions isn’t significant (if they cover the same domain/support of the function being approximated); **2.**  Critically, shifted basis functions ( (\( I_i \))) reconstruct non odd-sym functions (Fig. 1C–F & SI-1) – irrespective of the distribution from which they’re drawn.
>
> 2.**Training time BPTT**:  We’ve **added tables comparing training time across models & tasks (SI-F & Table 1,2)**.
>
> 3.**Lorenz Attractor**: We’ve **added a figure (SI-7)** showing a sample true & fitted trajectory & its corresponding time-series.
>
> 4.**Interpretability**: We thank the reviewer for this question, as this is a key contribution of our paper. Our work shows: a low-rank RNN implementing a particular dynamical system can be seen as providing a description of an ODE using a linear combination of nonlinear basis functions; the role of each neuron in a low-rank RNN is thus to contribute a single basis function. This perspective removes the need to use dimensionality reduction methods on trained RNNs (e.g., PCA, which other methods use to interpret recurrent connectivity/dynamics), or numerical optimization methods to find fixed points (common approach to interpret full rank RNNs). Thus, to summarize the key features:
>    - **1.** Each neuron represents a specific non-linear basis
>    - **2.** The ODE is a weighted sum of each basis, showing each neurons contribution
>    - **3.** OMP selects optimal neuron profiles for approximating a function, creating smaller networks
>    - **4.** Time-varying dynamics map directly to time-varying basis functions (or neuron responses).
>
> We’ll edit the text (Sec. 3 & 7) to clarify these points.
>
> **Real World Applicability**: We build on literature on “task training” RNNs (“weaknesses”- a. rank r - 2: real world datasets), which focus on training models to perform these “simple” (i.e rank 1 or 2) neuroscience tasks (eg. decision-making: Section 6 of our paper). However, to the reviewer’s comment, we’ve **added two additional dynamical portraits representing evidence accumulation (Fig SI-5)**.

---

> > ### Author Response · Authors · 2024-11-29
> >
> > Dear Reviewer RBkH
> >
> > Thank you once again for your initial review of our paper. As the discussion deadline approaches, we wanted to kindly follow up to ensure you are satisfied with our responses and the revised manuscript, or to see if there are any remaining concerns that need to be addressed.
> >
> > We have carefully addressed all the points raised in your feedback regarding the weaknesses and believe that our latest revision offers a clearer presentation of our work (along with extended simulations to address your concerns). We would greatly appreciate any additional feedback you may have. Thank you again for your time and invaluable insights!

---

### Meta-Review · Area_Chair_iH5D · 2024-12-20

**Metareview:**

The authors present a method for analyzing and training low-rank recurrent neural networks. Many of the authors had concerns about the presentation of the paper, as well as the novelty of the results. Those that engaged in discussion were generally persuaded that the authors improved the paper during the discussion period, but no one was persuaded to provide more than weak support for the paper. Given these concerns, I recommend against acceptance.

**Additional Comments On Reviewer Discussion:**

Although one reviewer who gave a very low score did not engage in the discussion period, they would have had to raise their score from a 3 to a 6 during the discussion period to go from reject to accept, which seems unlikely.

---

### Decision · Program_Chairs · 2025-01-22

Reject